# Measurement of the Berry curvature of solids using high-harmonic spectroscopy

Tran Trung Luu [1] & Hans Jakob Wörner[1]

Berry phase and Berry curvature have become ubiquitous concepts in physics, relevant to a variety of phenomena, such as polarization, various Hall effects, etc. Studies of these phenomena call for characterization of Berry phase or curvature which is largely limited to theory, and a few measurements in optical lattices. In this work, we report polarimetry of high-harmonic emission from solids and exploit this novel capability to directly retrieve the Berry curvature of $\alpha$-quartz. We show that the two manifestations of broken inversion symmetry in solids lead to perpendicular or parallel polarization of even harmonics with respect to the driving field. Using semiclassical transport theory, we retrieve the Berry curvature from spectra measured in perpendicular polarization, the results being supported by ab initio calculation. Our work demonstrates an approach for the direct measurement of Berry curvature in solids, which could serve as a benchmark for theoretical studies.

[1] Laboratorium für Physikalische Chemie, ETH Zürich, 8093 Zürich, Switzerland. Correspondence and requests for materials should be addressed to T.T.L. (email: trung.luu@phys.chem.ethz.ch)

**I**n his seminal work, Berry[1] showed that when a quantum system evolves adiabatically in a closed circuit $C$ in a parameter space R, it will acquire a geometrical phase change $\gamma_n(C)$ additionally to its intrinsic dynamical phase $-iE_nt/\hbar$, where $|n(R)\rangle$ is an eigenstate of energy $E_n$ of Hamiltonian H. The phase $\gamma_n(C)$, later called Berry phase, can also be expressed in three-dimensional (3D) space as $\gamma_n(C) = -\iint_C d\mathbf{S} \cdot \mathbf{\Omega}_n(R)$ where $\mathbf{\Omega}_n(R)$ is a vectorial quantity known as Berry curvature. It is the source of the anomalous velocity[2], the intrinsic contribution to anomalous Hall effects[3–6]. It also plays a role in topological insulators[7] and other fields[8]. Therefore, the determination of the Berry curvature is of fundamental importance to condensed matter physics but has so far only been accessible to numerical calculations[6,9–12] or measurements in optical lattices[13,14].

One important aspect that accompanies the Berry phase in affecting optical and electronic properties of solids is the symmetry. Almost all physical and chemical processes are governed by selection rules that are direct consequences of symmetry principles[15]. Time-reversal symmetry in solids results in symmetric bandstructure[15], which is the reason behind the generation of only odd harmonics in light–matter interactions (Supplementary Note 1). Broken spatial inversion symmetry in solids gives rise to the Berry curvature being an antisymmetric function[8] in momentum space, which is one of the possible causes of the emission of even harmonics in high-order harmonic generation (HHG) from solids[16].

HHG in solids is an emerging attosecond technology[16–20], for which different physical mechanisms have been proposed: collective contributions of both intraband and interband excitations[18,21,22], only intraband currents[16,17,19,23], generalized Wannier–Stark ladder[24], or generalized re-collision model[25–27]. Although the main physical mechanisms remain controversial, the semiclassical transport theory has been shown to successfully explain both the spectral and temporal structures of high-harmonic emission from silicon dioxide ($SiO_2$)[19,23].

In this work, we introduce polarimetry of high-harmonic emission from solids and use it to perform a complete characterization of the vectorial properties of even- and odd-harmonic emission. We utilize this innovative capability to identify different manifestations of broken symmetry in HHG from $\alpha$-quartz. Although HHG from $\alpha$-quartz has been investigated very recently[28,29], characterizing broken symmetry and its application in extracting electronic properties of $\alpha$-quartz has not been performed. Furthermore, we make use of extended semiclassical transport theory[8,30] to retrieve the Berry curvature, reporting what we believe to be the first measurement of this fundamental property of solids.

## Results

### Generation of extreme ultraviolet even harmonics.
Breaking of inversion symmetry in crystals results in a nonvanishing Berry curvature[8], which therefore manifests itself through the generation of even harmonics. Thus, we start our discussion by comparing high-harmonic spectra of non-inversion symmetric single-crystal z-cut $\alpha$-quartz (20 μm) with those generated in polycrystalline $SiO_2$ (100 nm) and completely amorphous fused silica (50 μm) in Fig. 1. We measure the spectral response of these samples at maximal non-damaging intensities, which are set to 90% of the intensities where first damages are observed. The crystalline $\alpha$-quartz sample has the thinnest commercially available thickness. The thicknesses of the other $SiO_2$ samples have been chosen to assess the roles of macroscopic effects and to serve as a reference for the $\alpha$-quartz sample.

The polycrystalline $SiO_2$ and the fused-silica sample show very similar spectral characteristics (red and orange lines) consisting of

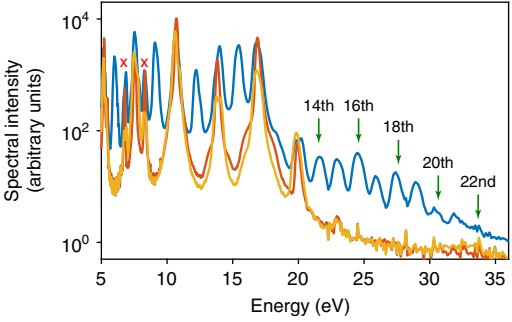

**Fig. 1** High-harmonic generation from various forms of $SiO_2$. Measured spectra generated from the crystalline $\alpha$-quartz (solid dark blue line), polycrystalline $SiO_2$ (solid red line), and fused silica (solid orange lines) driven by linearly polarized 30 fs pulses centered at 800 nm. All measurements are performed at room temperature (300 K). Each sample is exposed to the maximal non-damaging intensity ($\approx 1.0$ V/Å—$3.8 \times 10^{13}$ W/cm²—in all three cases). Clear even harmonics can be seen up to 34 eV (22nd harmonic order) in the emission from the $\alpha$-quartz. Red crosses denote the second-order diffraction from the grating-based spectrometer

only odd harmonics ranging from 5 to 23 eV at the peak field strength of $\approx 1.0$ V/Å ($3.8 \times 10^{13}$ W/cm²) inside the samples. By comparing the spectra of polycrystalline $SiO_2$ and fused silica, we extract two conclusions from their similarity. First, if the sample is not completely crystalline, the degree of crystallinity does not cause substantial differences in the HHG spectra. Second, the macroscopic effects due to propagation in different thicknesses are not severe in our experiment. In contrast to these similarities, HHG from $\alpha$-quartz is remarkably different. The cutoff photon energy in the $\alpha$-quartz spectrum is much higher than in fused silica, reaching up to 34 eV (22nd harmonic order). In addition, the $\alpha$-quartz spectrum exhibits even harmonics with spectral intensities comparable to odd harmonics. The first feature might be understood intuitively as follows: in a non-perfectly crystalline sample, which is made up from unit cells of variable sizes arranged in a distribution of orientations, the hopping of field-driven electrons in real space and the coherent build-up of photons depend strongly on these properties of the crystalline units. Since the emission of high-energy photons is more sensitive to the phase matching than low energy photons, constructive build-up of high-energy photons is much harder to achieve in a non-perfectly crystalline sample. In addition, although the existence of the second plateau in the $\alpha$-quartz spectrum could not be explained yet using our current understanding (Supplementary Note 4), the above observations already have one important implication for laser physics and/or solid-state photonics: it is possible to generate high-energy photons (reaching 34 eV) from $\alpha$-quartz using the widely used high-power Ti: Sapphire 30 fs laser pulses at 800 nm carrier wavelength. Furthermore, the above measurements show that the primary manifestation of broken symmetry in single-crystal $\alpha$-quartz is the emission of even harmonics. Our resultant spectrum is qualitatively similar to the case of isotropic media[31] where the electric field is the source of symmetry breaking. This interesting fact (explained in detail in Supplementary Note 1) shows that regardless of the exact cause of broken inversion symmetry (medium or electric field), the polarization response will include both even and odd harmonics, although with different spectral intensities.

Studies of the origin of even-harmonic generation from solid media through interaction with intense laser pulses could shed light on the electronic properties of the media. In the non-perturbative regime of our work (the peak intensity is on the

order of $10^{13}$ W/cm² or higher), even harmonics could be generated due to nonvanishing Berry curvature as discussed in ref.[16] or by quantum interference of the direct and indirect excitation pathways due to strong coupling between the valence or conduction bands[18,21,32] or simply asymmetric dipole element[33]. The straightforward consequence of these different mechanisms is that the emitted even harmonics should be polarized perpendicular to the incident electric field polarization in the first case and parallel in the latter cases. Although recently investigated in the terahertz regime[32], polarimetry of HHG from solids in the extreme ultraviolet (EUV) range has not been reported prior to this work.

**Polarimetry of EUV HHG from solids**. As a next step to comprehensively measure polarization-resolved HHG from solids, we built a broadband all-reflective EUV polarizer (Supplementary Note 2). The combination of this EUV polarizer, its rotary stage, and a rotatable sample mount (see Fig. 2a) allows us to characterize the amplitude and polarization of the emitted HHG spectra at an arbitrary orientation of the sample.

We first study the effect of crystal orientation on the HHG spectra at a given polarization. Therefore, the polarizer is fixed such that it allows the transmission of polarization parallel to the polarization of the incident electric field, which is also fixed. In this configuration, only the parallel polarization component is recorded, whereas the perpendicular polarization component is left undetected. The measured spectra for different rotation angles

(Fig. 2a) of the sample are shown in Fig. 2d. In these measurements, the maximum electric field strength was reduced to permit long-term measurements without damaging the sample. Therefore, the cutoff photon energy is reduced compared to Fig. 1. The calibration of the exact sample orientation angle is done utilizing the relative spectral intensity of different crystal directions within the semiclassical model. The following features can be observed: first, both odd and even harmonics are recorded with the even harmonics being slightly weaker than the odd harmonics in this energy range, which is in agreement with Fig. 1. Second, both odd and even harmonics are modulated with the periodicity of 60 degrees, revealing the six-fold symmetry of the Brillouin zone for the z-cut projection (Fig. 2c). Third, the modulation depth of the odd harmonics is much weaker when compared to the modulation depth of the even harmonics, which means that the linearized projection of the even harmonics is much more dependent on crystal orientation than the odd harmonics. Finally, the above measurements show that the even harmonics display maximal intensities when the driving field points along the Γ − K direction, whereas the odd harmonics are maximized along the Γ − M direction.

Having gathered the dependence of the emitted HHG spectra on the crystal orientation, we proceed to characterize and assign the manifestations of broken symmetry in α-quartz. We do this by performing two polarization measurements by rotating the polarizer, while the α-quartz crystal is oriented such that the laser polarization is parallel to either the Γ − K or Γ − M direction. These remarkably different results are shown in Fig. 2e, f,

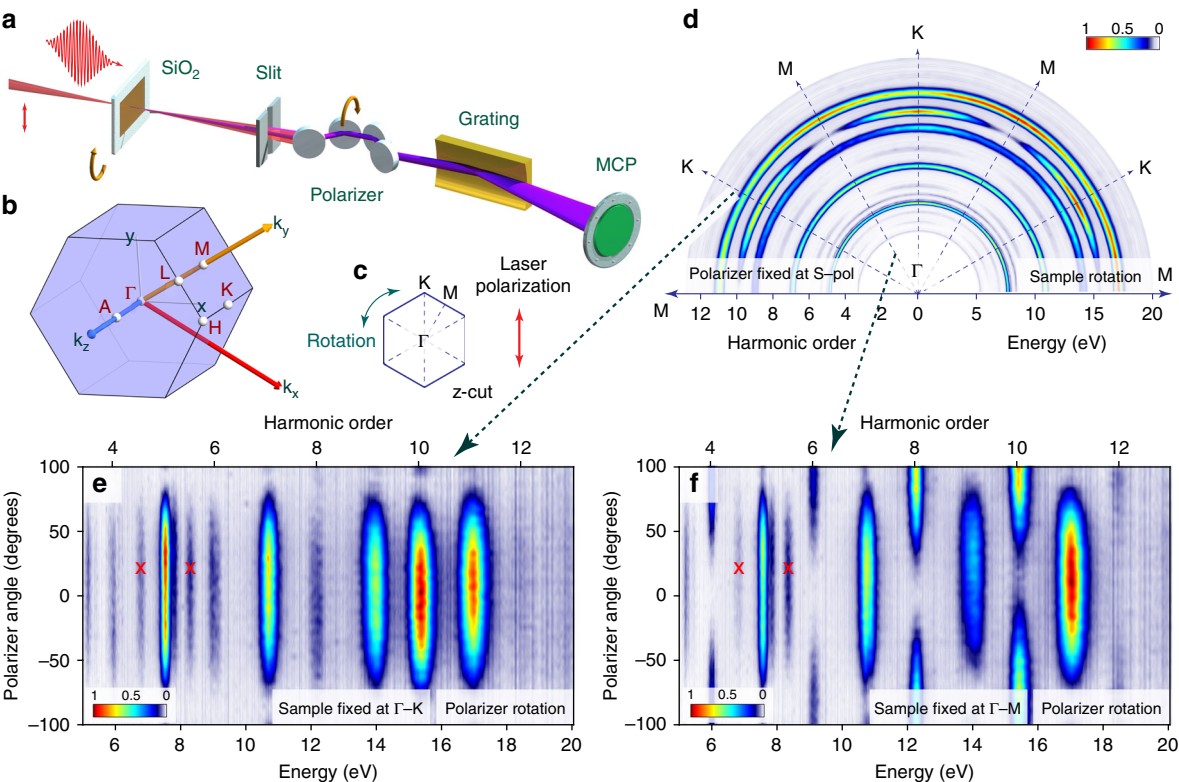

**Fig. 2** Unveiling physical mechanisms of even-harmonic generation from the α-quartz. **a** Experimental apparatus showing the spectral measurement with rotatable target and EUV polarizer. **b** Perspective view of the Brillouin zone of α-quartz showing both the Cartesian coordinates, as well as **k**-vectors. **c** Projection of the Brillouin zone for a z-cut α-quartz crystal. **d** Recorded spectral intensity (false-color) as a function of the rotation angle of the α-quartz sample. The polarizer is fixed at S polarization. The incident electric field is a linearly polarized 30 fs pulse at a carrier wavelength of 800 nm. **e**, **f** Recorded spectra as a function of the rotation angle of the polarizer, when the α-quartz sample is oriented at the Γ − K and Γ − M directions, respectively. Zero degree corresponds to the polarization direction parallel to the incident linearly polarized electric field. Red crosses denote the second-order diffraction from the grating-based spectrometer

respectively. The general dissimilarity shows that broken symmetry has two distinctively different manifestations along these two directions. Along the $\Gamma - K$ direction, both odd and even harmonics are recorded with odd harmonics dominating in intensity over even harmonics (with the exception of the 10th harmonic). Because both even and odd harmonics are linearly polarized and parallel to each other (Supplementary Note 2) and the incident electric field, even harmonics recorded in this direction can be assigned to quantum interference of direct and indirect excitation pathways included in the semiconductor Bloch equations (SBEs) for multiple bands[18] or asymmetric dipole element. The semiclassical transport theory in its plain form cannot explain the observation made for this particular direction (with a small exception described in Supplementary Note 1). Furthermore, the spectral intensity of the odd harmonics is much higher than the even harmonics at low photon energies, whereas the intensities are very similar at high photon energies. A similar observation has been made in monolayer molybdenum disulfide[16], also possessing a hexagonal lattice. This spectral evolution of the ratio of odd/even harmonic spectral intensity in $\alpha$-quartz lends further support to the interference picture. Quantum interference of direct and indirect excitation pathways is only enhanced in regions of couplings between the interband (valence ↔ conduction) transitions, leading to strong even harmonics at energies higher than that of the bandgap, which amounts to $\approx 9$ eV.

A remarkably different manifestation of broken symmetry can be seen in the $\Gamma - M$ direction as shown in Fig. 2f. All odd and even harmonics are emitted with similar spectral intensities, but perpendicular polarizations. A careful analysis of the contrast confirms that the odd harmonics are linearly polarized, parallel to the incident electric field polarization. The even harmonics are also linearly polarized, but in the direction perpendicular to the polarization of the odd harmonics. Because the even harmonics are only emitted at certain orientations, while the odd harmonics are emitted at all orientations, this observation explains the higher (nearly perfect) contrast of linearized projection of even harmonics compared to odd harmonics that we observed in the orientation-dependent spectra (Fig. 2d). Furthermore, in the $\Gamma - M$ direction, because the even-harmonic emission is polarized perpendicular to the incident electric field, it could only be generated due to nonvanishing Berry curvature in the bands.

## Discussion

In the next step, we take advantage of these comprehensive characterizations of the HHG spectrum and polarization and invoke the extended semiclassical transport theory to the first order of the electric field to determine the Berry curvature of $\alpha$-quartz. This is the optimal approach, considering that numerical solution of the complete 3D SBEs with ab initio electronic properties would require significant efforts, while it would not allow us to extract physical properties as elegantly as the Berry phase formalism. Here, the $\mathbf{k}$-dependent group velocity $\mathbf{v}_\nu(\mathbf{k})$ can be expressed as [8, 30]

$$\mathbf{v}_\nu(\mathbf{k}) = \frac{1}{\hbar}\frac{\mathcal{E}_\nu(\mathbf{k})}{d\mathbf{k}} - \frac{e}{\hbar}\mathbf{E}(t) \times \mathbf{\Omega}_\nu(\mathbf{k}), \quad (1)$$

where $\mathbf{E}(t), \mathcal{E}_\nu(\mathbf{k}), \mathbf{\Omega}_\nu(\mathbf{k})$ are the electric field, band dispersion, and Berry curvature vector of the band $\nu$. The time-dependent group velocity gives rise to an oscillating electric current and as a result, to the emission of photons. In this formalism, scattering effects and electronic excitations are ignored as the electronic wavepacket is assumed to exist already in the conduction (valence) band. The right-hand side of Eq. (1) consists of two terms, which lead to the generation of odd harmonics polarized

parallel to the driving field $\frac{1}{\hbar}\frac{\mathcal{E}_\nu(\mathbf{k})}{d\mathbf{k}}$ and the even harmonics polarized perpendicular to the driving field $-\frac{e}{\hbar}\mathbf{E}(t) \times \mathbf{\Omega}_\nu(\mathbf{k})$. Because we are able to completely measure the spectra and their respective polarizations, we can directly associate the spectrum measured at the perpendicular polarization $S(\omega)_\perp$ to the Berry curvature of all involved bands $\nu$ as $\left|\mathfrak{F}\left[\sum_{\nu=1}^\infty \mathbf{E}(t) \times \mathbf{\Omega}_\nu(\mathbf{k})\right]\right|^2$. However, within the scope of this semiclassical theory, we assume that the total response of the Berry curvature vectors from all involved bands can be replaced by $S(\omega)_\perp \propto \left|\mathfrak{F}[\mathbf{E}(t) \times \mathbf{\Omega}_{\text{effective}}(\mathbf{k})]\right|^2$ where the $\mathbf{\Omega}_{\text{effective}}(\mathbf{k})$ is mainly contributed by the first conduction band. This equation, together with a precise determination (Supplementary Note 5) of $S(\omega)_\perp$, $\mathbf{E}(t)$, enables the retrieval of $\mathbf{\Omega}_{\text{effective}}(\mathbf{k})$. To facilitate the retrieval, taking into account the fact that $\mathbf{\Omega}_\nu(\mathbf{k})$ is an odd function of $\mathbf{k}$ due to the broken inversion symmetry[8] of $\alpha$-quartz, and the Fourier decomposition, we use

$$S(\omega)_\perp \propto \left|\mathfrak{F}\left[\mathbf{E}(t) \cdot \sum_{n=1}^\infty \gamma_n \sin(n\mathbf{k}(t)a)\right]\right|^2, \quad (2)$$

where $a$ is the lattice constant and $\{\gamma_n\}$ are the Fourier coefficients of the effective Berry curvature.

In order to reliably retrieve $\mathbf{\Omega}_{\text{effective}}(\mathbf{k})$ using Eq. (2), we perform intensity-dependent spectral measurements when the crystal is oriented in the $\Gamma - M$ direction, without the polarizer to increase the dynamic range of our measurements. At this particular orientation, as discussed above, even harmonics are emitted only in the perpendicular polarization; thus the polarizer is no longer needed in these measurements. With the comprehensive data recorded (Fig. 3a), we retrieve the Berry curvature $\mathbf{\Omega}_{\text{z,effective}}(\mathbf{k})$ as illustrated in Fig. 3b using standard minimization techniques. Since different Fourier coefficients of the Berry curvature contribute to different sets of high harmonics (Supplementary Note 5), in this case $E_{\text{peak}} \approx 0.8$ V/Å ($2.5 \times 10^{13}$ W/cm²), $a \approx 4.9$ Å, the meaningful spectral range of 5–20 eV results in a reliable retrieval of $\gamma_1-\gamma_{10}$ albeit other $\gamma_n$ could also be retrieved, but with lower accuracy. The highest photon energy considered (20 eV) is defined by our polarizer whereas, in fact, the measured intensity-dependent spectra extend significantly beyond 20 eV. Furthermore, the measurement of polarization-resolved intensities leaves the absolute sign of the Berry curvature undetermined. Because different sets of Fourier coefficients could lead to similar fitting results, we describe the retrieved Berry curvature by the probability density shown in Fig. 3b after using a large number of input values in the fitting procedure (Supplementary Note 5).

Although we cannot retrieve the absolute amplitude of the Berry curvature with high accuracy, its order of magnitude can be estimated by considering Fig. 2f; the even and odd harmonics have very similar spectral intensities. In semiclassical theory, this translates into an upper limit for the amplitude of the Fourier coefficients of band dispersion and curvature: $|\epsilon_n|na \approx E_0|\gamma_n|$ where $E_0$ is the maximum electric field strength and $\{\epsilon_n\}$ are the Fourier coefficients of the band $\nu$ which is defined through $\mathcal{E}_\nu(\mathbf{k}) = \sum_{n=0}^{n_{\max}} \epsilon_{\nu,n} \cos(n\mathbf{k}a)$.

To evaluate the significance of the retrieved $\mathbf{\Omega}_{\text{z,effective}}(\mathbf{k})$ and considering the fact that there is no existing calculation of Berry curvature of $\alpha$-quartz to the best of our knowledge, we have performed extensive density-functional-theory (DFT) calculations (Supplementary Note 3) utilizing existing methodologies[8,11,12,34,35]. The results show that the retrieved Berry curvature $\mathbf{\Omega}_{\text{z,effective}}(\mathbf{k})$ agrees relatively well with the calculated value for the first conduction band $\mathbf{\Omega}_{\text{z,25}}(\mathbf{k})$ (Supplementary Fig. 6). We note that the discrepancy between Berry curvatures calculated using different exchange-correlation functionals is remarkably large (Supplementary Fig. 5). Our results indicate that

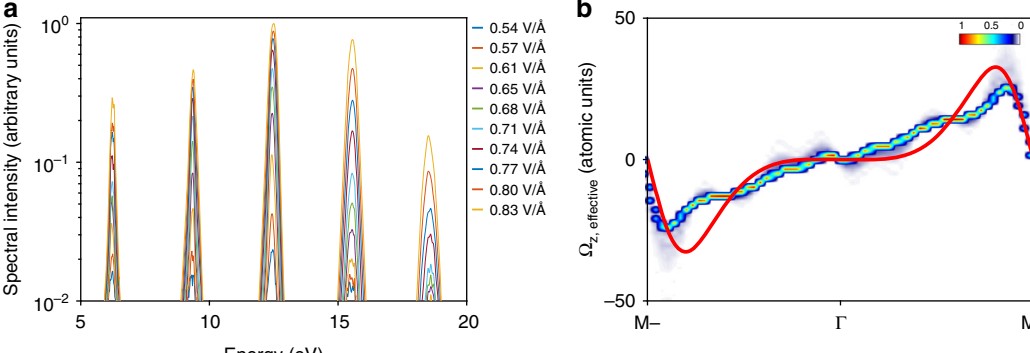

**Fig. 3** Retrieval of Berry curvature in single-crystal $\alpha$-quartz. **a** Spectra of even harmonics recorded through the intensity-scaling measurement when the crystal is oriented such that the laser polarization is parallel to the $\Gamma - M$ direction. Each spectrum is an average of four measurements under identical experimental conditions with an integration time of 10 s. For reasons of clarity, the raw data are plotted. The legends are the estimated peak electric field strengths inside the crystal, ranging from 0.54 to 0.83 V/Å or $1.1-2.7 \times 10^{13}$ W/cm$^2$. **b** Probability density of Berry curvature retrieved from the fitting. The absolute magnitude has been determined using the formula $|\epsilon_n| na \approx E_0 |\gamma_n|$ and should only be taken as an estimation of its order of magnitude. The solid red line is the calculated Berry curvature of the first conduction band using the direct approach (Supplementary Equation 28) based on DFT calculations using the MGGA functional with the modified Becke–Johnson exchange potential

the meta-generalized gradient approximation (MGGA) and modified Becke–Johnson potential provides the best description of the Berry curvature.

We have demonstrated HHG from the interaction of linearly polarized driving pulses with SiO$_2$ samples of three different degrees of crystallinity. The HHG spectrum from $\alpha$-quartz reaches out to much higher photon energies than those recorded from amorphous and polycrystalline samples and it consists of both even and odd harmonics with similar spectral intensities. Our measurements demonstrate the possibility of generating high-energy photons (extending up to 34 eV or the 22nd harmonic order) from standard Ti:Sapphire amplified 30 fs laser pulses. By constructing an all-reflective broadband EUV polarizer and rotating both the sample and the polarizer, we were able to perform the complete polarimetry of HHG from $\alpha$-quartz at arbitrary crystal orientations. The results show the two manifestations of symmetry breaking in solids, corresponding to different physical mechanisms: non-vanishing Berry curvature for the $\Gamma - M$ direction and the quantum interference of excitation pathways or asymmetric dipole moment for the $\Gamma - K$ direction. This approach naturally extends to more comprehensive studies of HHG of solids in other spectral ranges (e.g. infrared, THz). By utilizing the semiclassical transport theory including the anomalous velocity, we demonstrated a direct retrieval of the Berry curvature of solids. First DFT calculations of the Berry curvature of $\alpha$-quartz have been performed and the results show a near-quantitative agreement between the retrieved Berry curvature and the calculated Berry curvature of the first conduction band. To contribute to the ongoing active discussion on the mechanisms underlying HHG from solids, our experimental and theoretical models support the dominance of intraband current in the first conduction band as the driving mechanism of HHG from $\alpha$-quartz. Although our theoretical model requires an adiabatic approximation to hold such that the existing formalisms of the Berry curvature could be utilized, the idea of the methodology should be extendable to more general cases. The simplicity of our methodology is also its power. Our methodology can enable a new class of measurements of Berry curvature in solids. This work can also serve as a benchmark for the further development of ab initio calculations[8–10,12], as well as stimulate applications of solid-state coherent EUV sources in condensed matter physics and ultrafast photonics.

**Data availability**. The data that support the findings of this study are available from the corresponding author upon reasonable request.

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

## Acknowledgements

It is our pleasure to thank Denitsa Baykusheva and Andres Laso for their contributions to the construction of the experimental apparatus, and Eduardo V. Castro, Martin Grad-hand, Stanislav Kruchinin for fruitful discussions. We gratefully acknowledge funding from an ERC Starting Grant (307270-ATTOSCOPE), the ETH Zurich Postdoctoral Fellowship Program (FEL-31 15-2), and the Marie Curie Actions for People COFUND Program. Computational resources provided by the ETH Zurich Euler cluster are acknowledged.

## Author contributions

H.J.W. and T.T.L. conceived and designed the experiments. T.T.L. carried out the experiments, and performed the analyses and calculations. H.J.W. and T.T.L. discussed the results and wrote the manuscript.

## Additional information

**Competing interests:** The authors declare no competing interests.

