## [Peer Review File · Nature Communications]

Reviewers' comments:

Reviewer #1 (Remarks to the Author):

This paper investigates even harmonics measured from alpha-quartz, focusing in particular on harmonics with perpendicular polarization to the driving laser field. The authors show that Berry curvature can be reconstructed from the measured harmonics, showing good agreement with ab initio calculations.

By using complete characterization of the polarization of even harmonic emission in solids, this work obtains the first momentum resolved measurements of the Berry curvature. This is likely to attract wide-spread interest from the condensed matter community, since Berry curvature plays a role in the anomalous Hall effect and a number of other important phenomena. I therefore find these results extremely interesting and deserving of publication in a high profile journal.

I do have a question about the compatibility of the present findings with a theoretical publication that calculated HHG in alpha-quartz using time-dependent density functional (TD-DFT) theory for similar laser parameters, published by Otake in PRB 94, 235152, 2016.

In particular, the present paper seems to be in direct contradiction to the results of the PRB in regard to whether interband or intraband oscillations make the dominant contribution to HHG. The PRB claims that: "Our simulations indicate that the interband interaction is the dominant process". On the other hand, the present manuscript states: "These results thus also support the dominance of intraband current in the first conduction band as the driving mechanism of HHG from alpha-quartz".

Given particularly that the author of the above mentioned PRB is quite well-respected for his TD-DFT simulations in solids, in collaborations with K. Yabana, could the authors suggest a reason for the apparent discrepancy?

Minor comment: It would be helpful if the authors gave the maximum driving laser intensity plotted in Fig. 3 in terms of W/cm^2 (in addition to the $V/\text{\AA}$).

To conclude, due to the widespread interest in the condensed matter community that the present measurements of the Berry curvature will likely generate, I will be happy to recommend publication after the above question is answered.

Alexandra Landsman

Reviewer #2 (Remarks to the Author):

The manuscript entitled “Measurement of the Berry curvature of solids using high-harmonic spectroscopy” by Luu and Woerner presents interesting work on high-harmonic generation in quartz up into the XUV spectral range, analyzed with polarization resolution with particular regard to the Berry phase resulting from a broken symmetry in one of their highly crystalline samples. As a reference, they present measurements on polycrystalline and amorphous samples, which in contrast do not display even-order harmonics due to an effective restoration of quasi-symmetry. For their analysis of the Berry band curvature, they use a phase-retrieval scheme to reconstruct the phase up to a sign. Given the fact that measurements of absolute phases are impossible in their setup, and, moreover, very hard in general in this spectral range, this procedure is certainly the best approach possible, and it delivers solid results well compatible with the presented theory.

Yet, I am skeptical concerning the conclusions of the authors. The interpretation of the data in terms of a Berry phase may make sense but there are alternative explanations which need to be ruled out by the authors. Therefore, while the experimental results are solid and the relevance of the Berry phase would be highly interesting to the HHG community, I cannot recommend publication of the manuscript with the current interpretation.

When comparing the HHG in different crystal directions and polarizations, the authors relate the presence or absence of the even-order harmonics in one given polarization direction to quantum interference, or the Berry curvature, respectively, by exclusion (page 7). However, it has recently been shown by Langer et al. [Nature Photonics 11, 227 (2017)] that in the hexagonal semiconductor gallium selenide, quantum interference of excitation pathways in combination with linear interference is responsible for the complex polarization patterns of HHG, which are identical to the patterns discussed in the present manuscript: parallel orientation for excitation along G-K, and perpendicular for excitation along G-M. It is not clear why this description, which is backed by a

microscopic theory, should not be applicable in the present case, and why the Berry phase should be responsible instead.

The authors should also explicitly state what generation mechanism they suppose for the HHG in their case, which is, by their implication, intraband currents driven coherently by the external laser field $E(t)$. Quantum interference of excitation pathways similar to Ref. 18, as mentioned by the authors, makes sense only in this context.

One further concern is the overlap of parts of the present manuscript and the contained claims with work very recently published by You et al. [Nature Communications 8, 724 (2017)]. In particular, You et al. have already shown high-harmonic generation in crystalline quartz vs. fused silica, underlining the aspect of the simplicity of generation of high-energy photons in solids using a very similar scheme based on a laser source centered at 1700 nm, with CEP control. Here, the authors could in particular shine novel light on the actual generation mechanism by testing the phase coherence of their lower-order harmonics using interference spectroscopy with, e.g., the second to fourth harmonic of their laser source, if possible (optional). In any case, this aspect is not new anymore and is thus overstressed.

Furthermore, You et al. have already discussed the presence or absence, respectively, of even-order harmonics, depending on the grade of crystallinity. Luu et al. also discuss this point but do not reference the work by You et al.

Moreover, here, the interpretation is that the grain size of the solid restricts the coherent electron propagation distance (page 4, top). However, while this may be the case to some extent, You et al. have shown specifically for 800-nm excitation that the excursion length is on the order of one unit cell, i.e., only relevant comparing completely amorphous to slightly polycrystalline materials. The authors should comment on this. It might also be possible that in their case, as far as symmetry of the solid is concerned, presence/absence of even-order harmonics is related to interference of the generated HH rather than interference of electronic motion.

More general remarks:

The references of this manuscript seem to have been picked at an earlier stage, such that the most recent but highly relevant works as the ones given above are not cited, which the authors should do.

Fig. 3a: A semi-logarithmic plot seems more appropriate to me since the low-intensity spectral lines are hard to discern.

Response to the referees

We would like to thank both referees for their valuable time in assessing our manuscript! We thank them for their great suggestions, in response to which we have made significant changes to our manuscript. To ease their reading, we have highlighted the additions/changes in our resubmission. While the referees raised a few concerns, we believe that we were able to address them properly and hope that the referees will find our answers satisfactory.

Referee #1

This paper investigates even harmonics measured from alpha-quartz, focusing in particular on harmonics with perpendicular polarization to the driving laser field. The authors show that Berry curvature can be reconstructed from the measured harmonics, showing good agreement with ab initio calculations.

By using complete characterization of the polarization of even harmonic emission in solids, this work obtains the first momentum resolved measurements of the Berry curvature. This is likely to attract wide-spread interest from the condensed matter community, since Berry curvature plays a role in the anomalous Hall effect and a number of other important phenomena. I therefore find these results extremely interesting and deserving of publication in a high profile journal.

We thank the referee for your comments. We are very happy to hear that the referee finds our work interesting and considers it to be highly relevant to the condensed-matter community.

I do have a question about the compatibility of the present findings with a theoretical publication that calculated HHG in alpha-quartz using time-dependent density functional (TD-DFT) theory for similar laser parameters, published by Otake in PRB 94, 235152, 2016.

In particular, the present paper seems to be in direct contradiction to the results of the PRB in regard to whether interband or intraband oscillations make the dominant contribution to HHG. The PRB claims that: "Our simulations indicate that the interband interaction is the dominant process". On the other hand, the present manuscript states: "These results thus also support the dominance of intraband current in the first conduction band as the driving mechanism of HHG from alpha-quartz".

Given particularly that the author of the above mentioned PRB is quite well-respected for his TD-DFT simulations in solids, in collaborations with K. Yabana, could the authors suggest a reason for the apparent discrepancy?

We are grateful to the referee for pointing out the direct contradiction of our results with those reported in Otake PRB 94, 235152, 2016. We would like to answer this excellent question as follows:

- In this work, Otake used the photon energy of 1.55 eV which can therefore be directly compared to experimental results [Nature 521, 498-502, 2015]. While Otake's work represents an impressive effort regarding numerical simulations of HHG from alpha-

quartz, the calculated spectrum [Fig. 4 in PRB paper] does not agree at all with the experimentally measured spectrum [Fig. 3 in Nature paper]. We would therefore argue that the theoretical results of Otake are not validated by the available experimental observations. Our conclusions, in contrast, are based on experimental observations and a semi-classical model with minimal assumptions.

- Experimental time-domain measurements of the emitted attosecond XUV pulses [Nature 538, 359-363, 2016] show that the temporal characteristics of HHG from SiO₂ are best explained by pure intraband current.
- Additionally, we believe that the referee would agree that the results from calculations depend substantially on the approximations and assumptions being used. This is particularly true for DFT calculations, which significantly depend on the nature of the employed exchange-correlation functional. We note that this strong dependence is also observed in our DFT calculations (see Supplementary Information). The work of Otake does not discuss this dependence, such that it might well be that different results would be obtained with a different functional.

Minor comment: It would be helpful if the authors gave the maximum driving laser intensity plotted in Fig. 3 in terms of W/cm² (in addition to the V/Å).

Thank you for this comment: we used V/Ångström because of the following reasons:

- There is a confusion in the ultrafast community regarding the peak intensity. Scientists used both cycle-averaged and instantaneous peak intensities, which differ by a factor of 2. Taking into account that the laser pulses are getting shorter, we think that using the instantaneous peak electric field is less ambiguous.
- Specifying electric field strength has the advantage that it implies a relative high error in determining the absolute intensity, which requires a considerable effort if one wants to take into account all sources of error, including propagation of error.

Nevertheless, following your suggestion, we added the instantaneous peak intensity (taking into account the refractive index of the medium) to our values.

Referee #2

The manuscript entitled “Measurement of the Berry curvature of solids using high-harmonic spectroscopy” by Luu and Woerner presents interesting work on high-harmonic generation in quartz up into the XUV spectral range, analyzed with polarization resolution with particular regard to the Berry phase resulting from a broken symmetry in one of their highly crystalline samples. As a reference, they present measurements on polycrystalline and amorphous samples, which in contrast do not display even-order harmonics due to an effective restoration of quasi-symmetry. For their analysis of the Berry band curvature, they use a phase-retrieval scheme to reconstruct the phase up to a sign. Given the fact that measurements of absolute phases are impossible in their setup, and, moreover, very hard in general in this spectral range, this procedure is certainly the best approach possible, and it delivers solid results well compatible with the presented theory.

Yet, I am skeptical concerning the conclusions of the authors. The interpretation of the data in terms of a Berry phase may make sense but there are alternative explanations which need to be ruled out by the authors. Therefore, while the experimental results are solid and the relevance of the Berry phase would be highly interesting to the HHG community (...)

We thank the referee for his/her supportive comments and for identifying the value of our work to the scientific community.

When comparing the HHG in different crystal directions and polarizations, the authors relate the presence or absence of the even-order harmonics in one given polarization direction to quantum interference, or the Berry curvature, respectively, by exclusion (page 7). However, it has recently been shown by Langer et al. [Nature Photonics 11, 227 (2017)] that in the hexagonal semiconductor gallium selenide, quantum interference of excitation pathways in combination with linear interference is responsible for the complex polarization patterns of HHG, which are identical to the patterns discussed in the present manuscript: parallel orientation for excitation along G-K, and perpendicular for excitation along G-M. It is not clear why this description, which is backed by a microscopic theory, should not be applicable in the present case, and why the Berry phase should be responsible instead.

We thank the referee for this excellent comment. The group in Regensburg in collaboration with the Marburg group has done extraordinary work on HHG from solids and in fact all of the previous publications of T. T. Luu on HHG from solids have benefited significantly from their discoveries. However, we may not always have the same perspective and this work is an example.

Before we go into details, please allow us to give a broader overview: both the semiconductor Bloch equations utilized in the above mentioned work, as well as the Berry-phase formalism stem from the same fundamental principle: the Schrödinger equation in solids, applied to a system with broken inversion symmetry. Therefore, the Berry phase concept is *ab initio*, *microscopic theory* by itself. Only its implementation in semi-classical theory is an approximation due to the assumptions of semi-classical theory itself.

While the SBE and Berry phase are two ways of describing HHG from solids in perpendicular polarization, we would like to emphasize that these two descriptions are not mutually exclusive but rather different mathematical formulations of common physical phenomena. In our work, we consciously chose the Berry-phase formalism for the following reasons:

- Historically, since the introduction of Berry phase [M. V. Berry, Proc. R. Soc. London Ser. A 392, 45 (1984)], there have been numerous works employing this concept, most notably the modern theory of polarization [R. D. King-Smith and D. Vanderbilt, PRB, 47, 1651-1654, 1993] where the Berry phase is included e.g. in the explanation and computation of the piezoelectric tensor of GaAs. Thus, the Berry phase concept is well suited for calculation of polarization response, especially utilizing semi-classical theory.
- The semiconductor Bloch equations, by definition, capture a vast variety of light-solid interactions. However, being a generic light-matter interaction model, the SBEs are not well suited for applications where Berry phase is known to dominate

which involves adiabatic transport [RevModPhys, 82, 1959, (2010)].

Additionally, the Berry curvature has been hinted at as the mechanism for even-harmonic generation in Ref. 16.

- A theory is especially valuable or important when it explains experimental results. However, it would be even more important to be able to use that theory to predict new experimental results and to benefit from using it. This is where the main difference between the SBE and the Berry-phase formalism lies in our case. In the work by Langer et al. [Nature Photonics 11, 227 (2017)], the authors have successfully reproduced their experimental observations using their theory. Nevertheless, application of such a theory to the determination of unknown physical properties has been limited to the suggestion of potential benefits. In our work, by using the Berry-phase concept, we have been able to perform the first experimental measurement of Berry curvature in a real solid and validate the measurement by comparison with the independently calculated *ab-initio* values of the Berry curvature.
- In the work by Langer et al. [Nature Photonics 11, 227 (2017)], it would be ideal to solve the SBEs completely in three dimensions, using *ab-initio* calculated, three-dimensional dipole-matrix elements. In this case, the calculated polarization should be identical to the results obtained from polarization calculations using the Berry-phase concept [R. D. King-Smith and D. Vanderbilt, PRB, 47, 1651-1654, 1993] in the weak field regime, if the same Bloch wavefunctions are used as a basis and electron scattering processes are neglected. However, the authors of Langer et al. implemented the SBEs in an approximate way by using a one-dimensional model with three independent electric field vectors which prevents a direct translation of their results into the Berry-phase formalism. On the other hand, considering that solving the complete 3D SBEs with *ab initio* electronic-structure properties would require extensive efforts, we believe that our approach, based on the Berry-phase formalism in a semi-classical framework is the most suitable approach because it does not only reproduce the experimental results but it also retrieves new physical information

The authors should also explicitly state what generation mechanism they suppose for the HHG in their case, which is, by their implication, intraband currents driven coherently by the external laser field $E(t)$. Quantum interference of excitation pathways similar to Ref. 18, as mentioned by the authors, makes sense only in this context.

We thank the referee for this suggestion. The main mechanisms of HHG from solids have been a topic of ongoing debates ever since its discovery: intraband current [Nature Physics, 7, 138-141 (2011); Nature Photonics, 8, 119-123 (2014); Nature, 521, 498-502 (2015); Nature, 538, 359-363 (2016); Nature Physics, 13, 262–265 (2017)], interband polarization [Nature 522, 462-464 (2015), Nature 522, 462-464 (2015), Nature, 534, 520-523 (2016), Nature Physics, 13, 345-349 (2017)], or generalized Wannier-Stark ladder [Phys Rev Lett, 113, 213901 (2014)], etc. The complexity of the topic is further illustrated by the fact that the same group of authors mentioned by the referee has supported different mechanisms in different publications (albeit using different solids): intraband current [Nature Physics, 7, 138-141 (2011); Nature Physics, 13, 262–265 (2017)] and interband polarization [Nature, 534, 520-523 (2016), Nature Physics,

13, 345-349 (2017), Nature Communications 8, 724 (2017)]. Furthermore, although there have been two early publications on HHG from SiO₂ [Nature, 521, 498-502 (2015); Nature, 538, 359-363 (2016)] supporting the intraband picture, there is also the work on SiO₂ supporting the opposite picture [Nature Communications 8, 724 (2017)].

Therefore, considering all of the above, we believe that the debate concerning the relative importance of interband and intraband contributions is not settled, and the answer might well depend on the case at hand. Instead, our experimental and theoretical results would very much support the intraband picture and it is this “support” that we intended to emphasize in our manuscript.

One further concern is the overlap of parts of the present manuscript and the contained claims with work very recently published by You et al. [Nature Communications 8, 724 (2017)]. In particular, You et al. have already shown high-harmonic generation in crystalline quartz vs. fused silica, underlining the aspect of the simplicity of generation of high-energy photons in solids using a very similar scheme based on a laser source centered at 1700 nm, with CEP control. Here, the authors could in particular shine novel light on the actual generation mechanism by testing the phase coherence of their lower-order harmonics using interference spectroscopy with, e.g., the second to fourth harmonic of their laser source, if possible (optional). In any case, this aspect is not new anymore and is thus overstressed.

Furthermore, You et al. have already discussed the presence or absence, respectively, of even-order harmonics, depending on the grade of crystallinity. Luu et al. also discuss this point but do not reference the work by You et al.

We thank the referee for these valuable comments. As the referee correctly pointed out, our manuscript has been written before the publication of You et al. [Nature Communications 8, 724 (2017)], i.e. in March 2017, which explains why certain very recent references are missing. We have added all of these missing references in the resubmitted version.

We will first comment on the apparent overlap between You et al. and our manuscript. We agree with the referee that both manuscripts study high-harmonic generation from fused silica and crystalline quartz. However, this is also where the similarity ends. The main differences are the following:

- While the authors of You et al. could reproduce their experimental results with a simple, gas-like model, we went one step further in using our experimental results to retrieve a physical quantity – the Berry curvature - that has not been measured before. Additionally our results are confirmed by *ab initio* calculations.
- Our work measured and quantified the manifestations of broken inversion symmetry in detail, whereas their work only mentioned it.
- Our work uses a different laser source, which is much more commonly available. Although mid-infrared high power lasers (used in their work) are getting more and more common, high power Ti:Sapphire lasers (used in our work) undoubtedly surpass the mid-IR lasers in popularity and capability at the

moment. Consequently, our work would initiate interest in a much wider community of scientists and engineers.

- Our results are demonstrated in a different energy range, most importantly the photon energy achieved in our work is much higher (34 eV compared to 25 eV). Taking into account the fact that our laser wavelength is much shorter and our pulses are longer, our results are counter-intuitive as compared to You et al. and the known scaling laws from HHG in gases and therefore remarkable.

Moreover, here, the interpretation is that the grain size of the solid restricts the coherent electron propagation distance (page 4, top). However, while this may be the case to some extent, You et al. have shown specifically for 800-nm excitation that the excursion length is on the order of one unit cell, i.e., only relevant comparing completely amorphous to slightly polycrystalline materials. The authors should comment on this.

We thank the referee for this insightful comment. The reasons why we kept this discussion short in the manuscript are as follows:

- First, as soon as we talk about “excursion length” or “propagation distance”, we automatically assume that we are considering electrons as classical particle. While this assumption appears to be justified by the enormous success of the three-step model for HHG from gases, this assumption should be taken very carefully in solids. First, considering crystalline solids where atoms are densely packed in a uniform manner, the Wannier function (Fourier transform of the Bloch wave function) typically spans a large number of unit cells. This means that even for a simple transition in k-space under a weak electric field, the electrons could hop to a distant unit cell in real space. Therefore, focusing on “the excursion length is on the order of one unit cell, i.e., only relevant comparing completely amorphous to slightly polycrystalline materials” might result in a misleading understanding. However, our comment in the manuscript “constructive build-up of high energy photons is much harder to achieve in a non-perfectly crystalline sample” should still remain generally true.
- Second, considering amorphous or poly-crystalline solids where the Bloch formalism is not applicable, one has no choice but to solve the Schrödinger equation for the whole macroscopic solid to obtain an exact representation of HHG. In this case, the wavefunctions, solution of the Schrödinger equation, will evidently span macroscopic lengths. Therefore, the discussion of the “excursion length” or “propagation distance” will be less relevant.

We have adapted our text to better reflect this discussion.

It might also be possible that in their case, as far as symmetry of the solid is concerned, presence/absence of even-order harmonics is related to interference of the generated HH rather than interference of electronic motion.

We thank the referee for this comment. We are not entirely sure that we correctly understood this comment. In the last sentence of the first paragraph on page 4, we briefly mention the equivalent effects of symmetry breaking by medium and the laser fields and discuss this point in detail in Section I.A.1 of the Supplementary Material.

More generally, it seems difficult to us to clearly delineate the two effects mentioned by the referee. Indeed, purely odd harmonics are generated when the electronic dynamics, and therefore the radiated high-harmonic fields are anti-symmetric from one half-cycle to the next. One can

view this phenomenon as destructive interference of even harmonics from the two consecutive half cycles and constructive interference of the odd harmonics, or, equivalently, as a symmetry property (i.e. the inversion anti-symmetry) of the electronic dynamics.

Similarly, when the inversion symmetry is broken, the interference in the high-harmonic fields emitted by consecutive half-cycles is neither perfectly constructive, nor perfectly destructive in both odd and even harmonics. Similarly, the electronic dynamics is no longer inversion-antisymmetric from one half-cycle to the next, which may also be phrased in terms of an interference of symmetric and antisymmetric electronic currents.

More general remarks:

The references of this manuscript seem to have been picked at an earlier stage, such that the most recent but highly relevant works as the ones given above are not cited, which the authors should do.

Fig. 3a: A semi-logarithmic plot seems more appropriate to me since the low-intensity spectral lines are hard to discern.

Thank you so much for your great suggestions!

We have added the missing references, including the two mentioned by the referee and two other recent ones [Nature Communications, 8, 1686 (2017); Nature Photonics, 11, 594-599 (2017)]. We changed the plot of Fig. 3a to semi-logarithmic as suggested by the referee.

Once again, we would like to thank both referees for their precious time! We hope that our revision is satisfactory and that the referees will support publication of our revised manuscript.

REVIEWERS' COMMENTS:

Reviewer #1 (Remarks to the Author):

The authors responded to the referees' comments carefully and in detail. The question of the generation mechanism (interband vs. intraband) in SiO₂ is not definitively resolved, as the authors themselves acknowledge on page 5 of their response.

Nevertheless, the ambiguity on this particular point, does not detract from the value of this manuscript. The ability to directly extract Berry curvature is valuable and deserves further study in follow-up publications, which can furthermore focus on excluding other mechanisms (such as the ones pointed out by Referee 2).

I therefore recommend this manuscript for publication in Nature Communications.

Reviewer #2 (Remarks to the Author):

In their revised manuscript, Luu et al. have extensively responded to my comments and answered all open questions satisfactorily. In my view, they have positively increased the scope of their work by detailing the current view of the community regarding HHG mechanisms, and the role of Berry curvature in this context. Furthermore, I deem their justification of their model fully appropriate and I have hence no further reservations. All other comments have been considered as well.

Their work will be highly interesting to the HHG community and beyond that to a broader audience due to the relevance of the more general concept of Berry curvature.

I strongly recommend publication of the manuscript.

Response to the referees

Referee #1

The authors responded to the referees' comments carefully and in detail. The question of the generation mechanism (interband vs. intraband) in SiO₂ is not definitively resolved, as the authors themselves acknowledge on page 5 of their response.

Nevertheless, the ambiguity on this particular point, does not detract from the value of this manuscript. The ability to directly extract Berry curvature is valuable and deserves further study in follow-up publications, which can furthermore focus on excluding other mechanisms (such as the ones pointed out by Referee 2).

I therefore recommend this manuscript for publication in Nature Communications.

Thank you for your comment and the recommendation. We have strived to bring the most comprehensive perspective on the current topic of HHG from solids. We hope that eventually we will reach converged opinion.

Referee #2

In their revised manuscript, Luu et al. have extensively responded to my comments and answered all open questions satisfactorily. In my view, they have positively increased the scope of their work by detailing the current view of the community regarding HHG mechanisms, and the role of Berry curvature in this context. Furthermore, I deem their justification of their model fully appropriate and I have hence no further reservations. All other comments have been considered as well.

Their work will be highly interesting to the HHG community and beyond that to a broader audience due to the relevance of the more general concept of Berry curvature.

I strongly recommend publication of the manuscript.

We thank the referee for support and the recommendation. In order to keep the literature up-to-date, we have additionally cited one more publication: Phys Rev B. 96, 053850, (2017) which changes neither major nor minor conclusions of our work.

Finally, we would like to thank both referees for their precious time! We are grateful to your excellent comments and the great support.